# Decoration of Vertically Aligned Carbon Nanotubes with Semiconductor Nanoparticles Using Atomic Layer Deposition

**DOI:** 10.3390/ma12071095

**Published:** 2019-04-02

**Authors:** Anna Szabó, László Péter Bakos, Dániel Karajz, Tamás Gyulavári, Zsejke-Réka Tóth, Zsolt Pap, Imre Miklós Szilágyi, Tamás Igricz, Bence Parditka, Zoltán Erdélyi, Klara Hernadi

**Affiliations:** 1Department of Applied and Environmental Chemistry, University of Szeged, H-6720 Szeged, Hungary; szabo.anna@chem.u-szeged.hu (A.S.); gyulavarit@chem.u-szeged.hu (T.G.); tothzsejkereka@chem.u-szeged.hu (Z.-R.T.); 2Department of Inorganic and Analytical Chemistry, Budapest University of Technology and Economics, Muegyetem rakpart 3., H-1111 Budapest, Hungary; laszlobakos@hotmail.com (L.P.B.); karda412@gmail.com (D.K.); imre.szilagyi@mail.bme.hu (I.M.S.); 3Nanostructured Materials and Bio-Nano-Interfaces Center, Interdisciplinary Research Institute on Bio-Nano-Sciences, Babeș–Bolyai University, Treboniu Laurian Str. 42, RO-400271 Cluj-Napoca, Romania; pzsolt@chem.u-szeged.hu; 4Institute of Environmental Science and Technology, University of Szeged, Tisza Lajos krt. 103, H-6725 Szeged, Hungary; 5Department of Organic Chemistry and Technology, Budapest University of Technology and Economics, Budafoki út 8. F. II Building, H-1111 Budapest, Hungary; igricz.tamas@gmail.com; 6Department of Solid State Physics, Faculty of Sciences and Technology, University of Debrecen, P.O. Box 400, H-4002 Debrecen, Hungary; parditka.bence@science.unideb.hu (B.P.); zoltan.erdelyi@science.unideb.hu (Z.E.)

**Keywords:** vertically aligned carbon nanotubes, atomic layer deposition, semiconductor particles, semiconductor/CNT forest nanocomposites

## Abstract

Vertically aligned carbon nanotubes (VACNTs or “CNT forest”) were decorated with semiconductor particles (TiO_2_ and ZnO) by atomic layer deposition (ALD). Both the structure and morphology of the components were systematically studied using scanning (SEM) and high resolution transmission electron microscopy (HRTEM), energy-dispersive X-ray spectroscopy (EDX), Raman spectroscopy, and X-ray diffraction (XRD) methods. Characterization results revealed that the decoration was successful in the whole bulk of VACNTs. The effect of a follow-up heat treatment was also investigated and its effect on the structure was proved. It was attested that atomic layer deposition is a suitable technique for the fabrication of semiconductor/vertically aligned carbon nanotubes composites. Regarding their technological importance, we hope that semiconductor/CNT forest nanocomposites find potential application in the near future.

## 1. Introduction

The vertically aligned carbon nanotube (VACNT), a sub-branch of the carbon nanotube family, is a relevant nanotechnology research topic nowadays. Its structure was synthetized by Li et al. for the first time in 1996 [1]. The VACNT are often called carbon nanotube forests in the literature and usually synthetized by applying the catalytic chemical vapor deposition (CCVD) technique [2]. The carbon nanotube forests differ from carbon nanotubes only in terms of orderliness, and CNT forests are fixed to a (conductive) substrate [3,4,5,6] which might result in direct electrical connection. The carbon nanotube forests are often used in electrical devices due to their electrical conducting properties and can be found in microelectromechanical devices [7] such as gas sensors [8], but can also be used in the preparation of nanocomposite systems [9], and have been extensively used for field emission applications, as well [10,11]. In the literature, several methods are already published for the preparation of ZnO [12] and TiO_2_ [13] composites, such as electrochemical [14], sputtering [15], and atomic laser deposition (ALD) [16]. However, in the case of the above-mentioned structure (VACNT), principally, the composite formed only on the outer surface of the CNT forests, while inner carbon nanotubes remained bare without any coverage. Only a few publications can be found where real composites were formed which are homogeneously composed of vertically aligned carbon nanotubes covered by inorganic layer [12,13,16,17,18]. Due to the potential technological importance of semiconductor/CNT forest composites, there is demand to develop an effective synthesis process for its production. Composite materials are often used such as electron emitters [19], nanotransistors [20], electrochemiluminescence [21], and sensors [22].

Atomic layer deposition (ALD) is a convenient chemical coating method for nanostructures. A wide range of materials can be deposited layer after layer, which enables the control of the thickness on the nanometer scale [23,24]. ALD was already used to deposit different metal oxides on single and multi-walled carbon nanotubes, such as TiO_2_, ZnO, ZrO_2_, and Fe_2_O_3_. These composites were tested in many applications, e.g., as transistors, photodetectors, electrodes or in photocatalysis [12,13,25,26,27,28].

The addition of carbon nanotubes to various semiconductors seems to be a suitable solution for the elongation of the lifetime of photogenerated e^−^/h^+^ pairs, thus preventing their recombination [29]. The effect of other carbonaceous materials were also investigated [30], and found that carbon nanotubes (CNTs) proved to have the best electron sink capabilities making it suitable to store and transport photogenerated electrons of semiconductor materials [31]. Applying various fabrication methods, many different types of metal oxides, such as TiO_2_ [32,33], SnO_2_ [34], Cu_2_O [35], CeO_2_ [36] and ZnO [37,38,39] were already deposited onto the surface of multi-walled carbon nanotubes (MWCNTs). Due to their outstanding electrical properties, large surface area, hollow structure, and adsorption sites, carbon nanotubes are approved components in nanocomposites which proved to be effective and very sensitive gas sensors [40,41].

Since 1972, the most investigated semiconductor has been titanium dioxide [42], especially its anatase phase. This material can be considered as an ideal photocatalytic candidate [43] with its band-gap (3.1–3.2 eV ≈ λ = 387–400 nm) being quite close to the visible light absorption range. Its absorption overlap with the sunshine radiation at the surface of Earth enabling the possibility to utilize sunlight [43]. Titanium dioxide is inexpensive and non-toxic, moreover, it is available in large amounts. Its exceptional properties combined with MWCNT offer promising materials for potential applications.

Due to its outstanding performance in electronics, optics, and photonics systems, ZnO also has widespread attention in the literature [44]. It is also applied as photocatalyst under UV irradiation (its band gap is 3.37 eV) for the degradation of various organic contaminants [40,45]. Among other semiconductor oxides, the utilization of ZnO as photocatalyst is favorable because of its relatively large quantum efficiency [46] and also its large exciton binding energy (60 meV) [47].

The CNT has beneficial properties, such as a large surface area, excellent electrical conductivity and a high capacity for electron reservoirs. Therefore, significant achievements have already been published regarding the improvement of the photocatalytic efficiency of ZnO/CNT [48] and TiO_2_/CNT [49] composites and to understand the mechanisms of enhancing the photocatalytic performance. Due to their prominent properties, these composites are often used in gas sensors, too, for e.g., TiO_2_/CNT for the detection of H_2_ [50] or ZnO/CNT for the detection of NH_3_ [51]. Many studies have been published in the literature regarding the synthesis methods of ZnO, including wet chemical-based methods [52], plasma-assisted sputtering [53], and microwave irradiation [54], while for TiO_2_ preparation usually sol-gel [55] and hydrothermal methods [49] are applied.

Summarizing the literature data for CNT forest nanocomposites, it can be concluded that the deposition of semiconductor materials into the intertubular region of CNT forests is rather challenging. Supposedly, due to poor wettability of the CNT surface, regular impregnation techniques are generally miscarried. In consequence, the aim of this work was to apply atomic layer deposition for the fabrication semiconductor/CNT forest nanocomposites using either Ti or Zn precursor.

## 2. Materials and Methods

### 2.1. Materials

During the experiments the following materials were used: Aluminum plates (WRS Materials Company, San Jose, CA, USA), cobalt (cobalt(II)-nitrate hexahydrate, 99% (Sigma-Aldrich, Saint Louis, MO, USA), iron (iron(III)-nitrate nonahydrate, 99.9% (Sigma-Aldrich, Saint Louis, MO, USA) precursors and absolute ethanol (VWR) were used in the formation of the catalyst layer. For the CCVD synthesis ethylene (purity > 99.9%), hydrogen (purity 99.5%), and nitrogen (purity 99.995%) gases were applied, which were supplied by the company of Messer Hungary (Szeged, Hungary). The precursors used during the atomic layer deposition were TiCl_4_ and Zn(C_2_H_5_)_2_ from Sigma-Aldrich.

### 2.2. Catalyst Preparation

First, iron and cobalt salts (0.888 g Fe(NO_3_)_3_·9H_2_O and 0.855 g Co(NO_3_)_2_·6H_2_O) were dissolved in 50 cm^3^ of absolute ethanol with the catalyst ratio of 2:3 and this catalyst ink had a transition metal concentration of 0.11 M. The effect of aging was avoided, the solution was prepared freshly before the dip-coating method. 

In the following step, the catalyst layer on aluminum sheet was prepared by applying the dip-coating technique. Before this, the substrate was washed with distilled water, ethanol, and acetone in order to remove all contaminations (motes, grease spots, etc.), then the sheet was cut into 3 × 2.5 cm sized pieces. Thereafter, the clean substrate was heat treated for 1 h at 400 °C in a static oven, and in this way–according to our former results–a thicker native oxide layer formed on the aluminum sheet which promoted the production of CNT forests during CCVD synthesis [3]. 

Dip-coating is an easy and well-controllable method to form a catalyst layer on the substrate. The aluminum sheet was submerged in the catalyst ink for 10 s, where the dipping and withdrawal speeds were 200 mm × min^−1^. The catalyst layer was built by KSV dip coater LM (KSV Instruments Ltd., Helsinki, Finland). The aluminum sheet with catalyst layer were heat treated again at 400 °C for 1 h in order to stabilize the catalyst on the surface of the substrate.

### 2.3. CCVD Synthesis

The CNT forests were prepared by the CCVD technique. The cut-to-size aluminum sheets were placed into a quartz boat and put in the pre-heated tube furnace at 640 °C (the aluminum melting point is 660 °C). The reaction time was 15 min, and the carbon source was ethylene (70 cm^3^/min), the carrier gas was nitrogen (50 cm^3^/min), the reducing agent was hydrogen (100 cm^3^/min), and the gas feed contained water vapor (30 cm^3^/min) in every experiment. 

During the first step of the CCVD synthesis, in the reactor, inert atmosphere was set by circulating nitrogen for 2 min. As the next step, the hydrogen gas valve was opened for 5 min, in order to reduce the catalyst particles. Thereafter, ethylene gas and water vapor were introduced into the system for 15 min. At the end of synthesis all gas flows were closed except for the nitrogen. After rinsing the reactor, the quartz tube was taken out from the furnace. The system was cooled down to room temperature, then the as-prepared samples were removed from the reactor. The structure of as-prepared vertically aligned carbon nanotubes is shown in Figure 1, with an average height of CNT forest of 15.5 µm. A typical specific surface area value for multiwalled carbon nanotubes is approx. 180 m^2^/g [33]. The estimated surface of CNT to be coated was a few cm^2^. 

### 2.4. Atomic Layer Deposition

Atomic layer deposition of the metal oxides was carried out at 1 mbar pressure in a Beneq TFS-200-186 ALD thermal reactor (Beneq, Espoo, Finland) equipped with a cross-flow reaction chamber with maximum reaction space diameter of 200 mm and height of 3 mm for fast film processing. TiO_2_ was deposited by the reaction of TiCl_4_ and H_2_O at 300 °C, and ZnO layers were made with (C_2_H_5_)_2_Zn and H_2_O at 200 °C. For TiO_2_, 400 cycles were used, whereas for ZnO, 120 were used; one cycle was composed of 0.3 s metal precursor pulse, 3 s N_2_ purge, 0.3 s H_2_O pulse and 3 s N_2_ purge. (Preliminary measurements revealed the sufficiency of even shorter pulse-purge periods; however longer cycles were applied for safety). The schematics of the ALD method are shown in Figure 2. Oxide layers were also grown on glass substrates under the same conditions and the theoretical thickness of the layers was measured by profilometer (AMBIOS XP-1, Crediton, UK); 40 nm for TiO_2_, and 20 nm for ZnO. The growth speed for TiO_2_ and ZnO was found to be 0.1 nm/cycle and 0.17 nm/cycle, respectively. The non-uniformity of as-prepared layers was also investigated by profilometric measurements on glass substrates arranged evenly on the entire plate of the reaction chamber with a diameter of 200 mm. Results revealed that non-uniformity on the entire plate was lower than 5% in the relevant thickness range. (Since our sample was much smaller than the plate, we can presume that non-uniformity is negligible in the current system.)

### 2.5. Characterization of Samples

Transmission electron microscopy (TEM) measurements were carried out using a FEI Tecnai G2 20 X-TWIN (200 keV) type instrument (FEI, Hillsboro, OR, USA), in order to test the quality of the carbon nanotubes. The samples were removed from the Al surface, then put into an Eppendorf vial and a small amount of the CNT was suspended in 1.25 cm^3^ absolute ethanol. In the next step, 2–3 droplets were dropped from this suspension onto the surface of a holey carbon grid (Lacey, CF 200, Electron Microscopy Sciences, Hatfield, USA. The TEM images were analyzed by using ImageJ software (Bethesda, MD, USA).

Scanning Electron Microscopy (SEM) analyses were performed with a Hitachi S-4700 Type II FE-SEM (5–15 keV) type instrument (Tokyo, Japan), in order to determine the orientation of the CNT forests. The samples were tilted at a 35° angle in all cases to investigate the carbon nanotube forests from each side. SEM images were analyzed by using ImageJ software. The above-mentioned instrument was also used to perform the EDX analysis, complemented by a Röntec XFlash Detector 3001 detector (Bruker, Karlsruhe, Germany).

Raman spectra were recorded on a Jobin Yvon Labram Raman instrument with an Olympus BX41 microscope (Tokyo, Japan) using green Nd-YAG laser (λ = 532 nm).

XRD patterns were recorded on a PANanalytical X’Pert Pro MPD X-ray diffractometer (Malvern Panalytical Ltd., Malvern, UK) using Cu Kα radiation.

## 3. Results

For the fabrication of metal oxide/CNT forest composites, both ZnO and TiO_2_ were deposited onto the surface of carbon nanotubes using the ALD technique. Samples were thoroughly characterized right after deposition, then both ZnO and TiO_2_ coated carbon nanotube forests were heat-treated in order to observe the change in the crystal structure of the semiconductor oxides. During heat treatment, inert atmosphere was used because carbon nanotubes burn away in the presence of oxygen atmosphere at 400 °C. The heat treatment was carried out in a tube furnace for 4 h, at 400 °C in argon atmosphere.

### 3.1. ZnO and TiO_2_ Coated Carbon Nanotube Forests

#### 3.1.1. Scanning Electron Microscopy Observations

SEM measurements revealed that the deposition of both metal oxides were successful. As can be seen in Figure 2, carbon nanotubes were decorated with nanosized particles in the whole bulk of the CNT forest. Analyzing the SEM images, it was found that the structure of the carbon nanotube forests was unmodified, and the TiO_2_ and ZnO were located on the carbon nanotube surface (Figure 3). As expected, no discernible changes were identified between the SEM images of the non-heat-treated and heat-treated samples. Therefore, the samples were investigated with EDX measurements too.

The average composition of the samples is shown in Table 1. While the pristine CNT forest was composed of nearly pure carbon, composite samples contained further elements. Besides the covering oxides (TiO_2_ and ZnO), traces of iron, cobalt and chlorine could be found in certain samples. The former two elements came from the catalyst of CNT growth, whilst, in the case of the TiO_2_/CNT, the chlorine was the residue from the TiCl_4_ precursor used during ALD. The presence of the aluminum substrate was also observable in these data; however, values were eliminated for better comparability. According to EDX results, more ZnO was deposited than TiO_2_. 

Interestingly, the relative amount of carbon compared to metal oxides (excluding aluminum support) became significantly lower after heat treatment, which may be due to the crystallization of TiO_2_. As was already mentioned, albeit the treatment was implemented under oxygen-free circumstances, a significant deficit of carbon was found compared to both Zn and Ti content.

#### 3.1.2. Raman Spectroscopy Results

Raman spectroscopy measurements were carried out in order to determine the Raman shift as well as to investigate the effect of TiO_2_ and ZnO on the Raman spectra of CNT forest.

On the Raman spectra of each sample (Figure 4), the D (~1340 cm^−1^) and G bands (~1580 cm^−1^) of the carbon are present, which come from the carbon nanotubes. The characteristic peaks of anatase can be seen on the spectrum of TiO_2_/CNT at 141, 400, 516 and 637 cm^−1^ [56]. In the case of ZnO/CNT, only two small peaks of the ZnO are visible, at 380 and 438 cm^−1^ [57] I_D_/I_G_ ratios of D and G peaks from carbon nanotubes were calculated to provide information on their graphitization (Table 2). The ratio of the D and G peaks intensities (I_D_/I_G_) changed during the deposition of the metal oxides (Table 2) which makes the chemical bond probable between Ti/Zn and CNT [58,59]. It was proved earlier that there is a strong correlation between the adherence of metal oxide particles to CNT surfaces and the density of defect sites in the carbon nanotube forest [60] Surprisingly, the heat-treated composite samples show lower I_D_/I_G_ ratios, thus fewer defect sites again. For example, in the case of ZnO/CNT sample I_D_/I_G_ was 1.08, an even better ratio than that of a pristine CNT forest and suggests only a few defect sites in the carbon nanotube. In spite of the presence of inert atmosphere during heat treatment, the oxygen in the sample could oxidize amorphous carbon and outer layers (which always contain more defects) of CNTs, resulting in increased graphitization values. This finding is in good agreement with observations done during the interpretation of EDX results.

#### 3.1.3. X-ray Diffraction Results

In Figure 5, the diffraction peaks of TiO_2_ could not be observed during the investigation of TiO_2_/CNT forest, because it was present only in a small amount on the surface of carbon nanotubes (see also the EDX data). The ZnO/CNT forest showed the peaks for the hexagonal ZnO (ICDD: 01-080-4199), and one peak (200) of the aluminum (ICDD: 00-004-0787) was present as well. A small diffraction peak is visible on the composites around 24° 2θ, which comes from the silicon sample holder.

#### 3.1.4. Transmission Electron Microscopy Observations

TEM measurements also confirmed that the metal oxides were successfully anchored onto the surface of carbon nanotubes composing forests. These measurements provided information about the quality of both carbon nanotubes and metal oxides as well.

From the TEM images (Figure 6) of composite samples, it can be concluded that the metal oxides truly enfold the surface of carbon nanotubes in both TiO_2_/CNT and ZnO/CNT composites. HR-TEM images also revealed that the number of walls in carbon nanotubes were 4–5 on average and their diameter varied between 5–6 nm, while the average diameters of the TiO_2_ and ZnO particles were 25 nm and 30 nm, respectively.

In the case of ZnO/CNT a more even coverage can be observed, compared to TiO_2_/CNT after heat treatment, but both metal oxides provided appreciable decoration on the CNT forest. From HR-TEM images, the interplanar lattice spacing was also calculated for metal oxides. It was observed that the lattice spacing did not change after heat treatment in both cases, thus no significant modification of crystal phases occurred. This value of 0.37 nm corresponds to the (001) facet of anatase in sample TiO_2_/CNT. At first glance, this value seems to be changed in the case of ZnO/CNT. Before heat treatment the lattice spacing was found to be 0.28 nm for ZnO, however, after heat treatment the lattice spacing was 0.32 nm based on HRTEM image. Literature data confirmed that both values belong to different facets ((0001) and (0110)) of a regular wurtzite hexagonal-structure ZnO particle [61,62]. It was also detected that the particles directly on the surface showed lower crystallinity compared to other quasi separated crystals.

## 4. Conclusions

Surveying the relevant literature, it became obvious that the deposition of different materials into the interior region of CNT forests is a real challenge. Probably due to wettability problems, conventional impregnation techniques are generally unavailable for this purpose [63]. Therefore, in this work, we applied atomic layer deposition for the fabrication semiconductor/CNT forest nanocomposites using either Ti or Zn precursor. The resulting composite materials were characterized by various methods such as SEM, XRD, HRTEM, XRD, and Raman spectroscopy. Characterization results revealed that the decoration of CNT forest with both Ti_2_O and ZnO was successful in the whole bulk of VACNTs. Knowing the properties of applied semiconductors, a follow-up heat treatment was applied, and structural change was detected. Comparing the characterization results, it was found that treatment at 400 °C caused major alteration in the carbon nanotube but not in the semiconductor properties. Regarding the technological importance of these semiconductor/CNT forest nanocomposites, the authors hope that their work contributes to the application of these materials in the field of photocatalysis, supercapacitors or other applications soon. 

## Figures and Tables

**Figure 1 materials-12-01095-f001:**
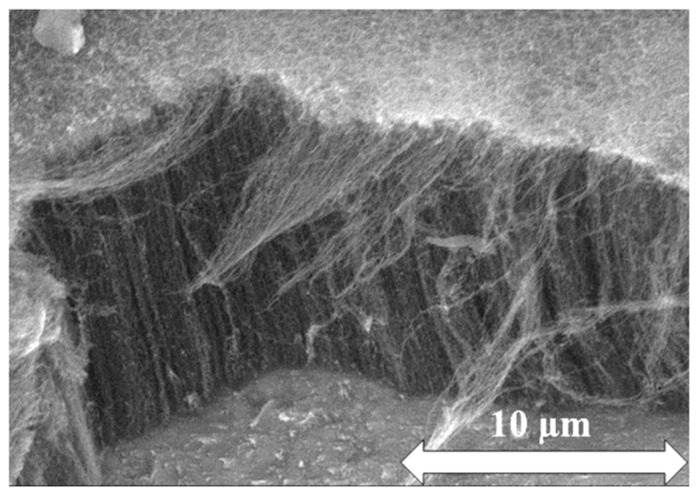
SEM image of as-prepared carbon nanotube forest.

**Figure 2 materials-12-01095-f002:**
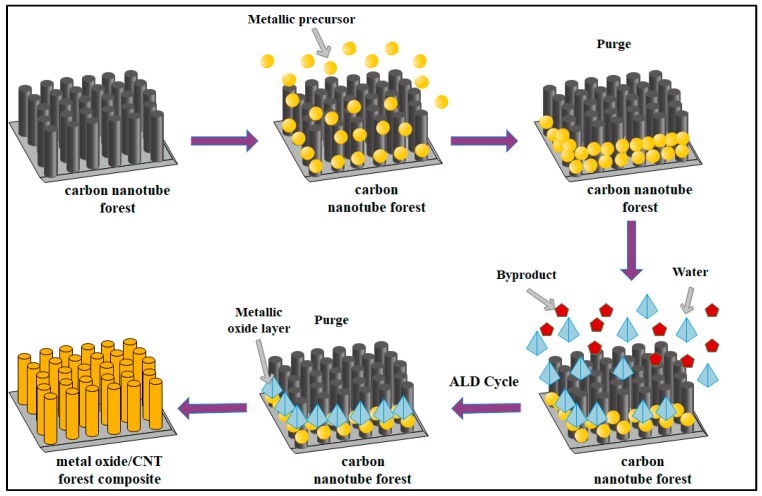
Schematic image of atomic layer deposition method.

**Figure 3 materials-12-01095-f003:**
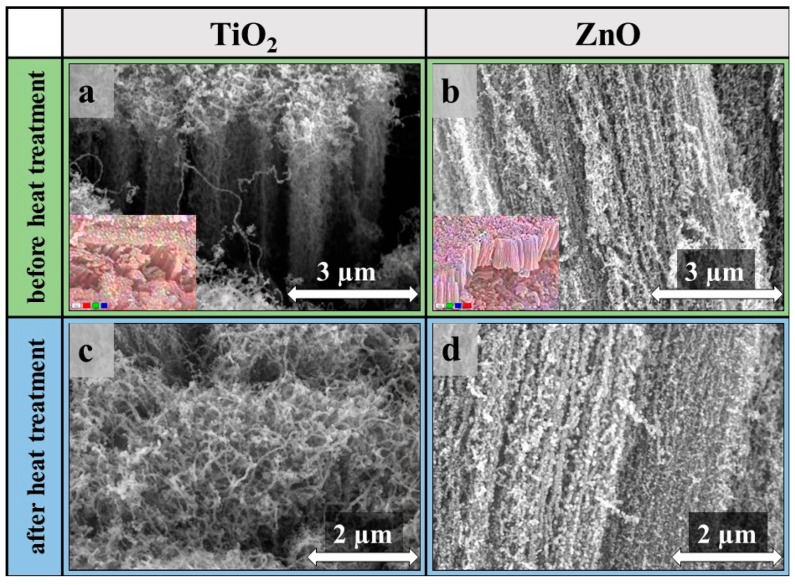
SEM images of TiO_2_ (**a**,**c**) and ZnO (**b**,**d**) coated CNT forests before (**a**,**b**) and after (**c**,**d**) heat treatment, element mapping of CNT forest composites (inset (**a**)–red: Ti, green: C, blue: O; inset (**b**)–red: Zn, green: C, blue: O).

**Figure 4 materials-12-01095-f004:**
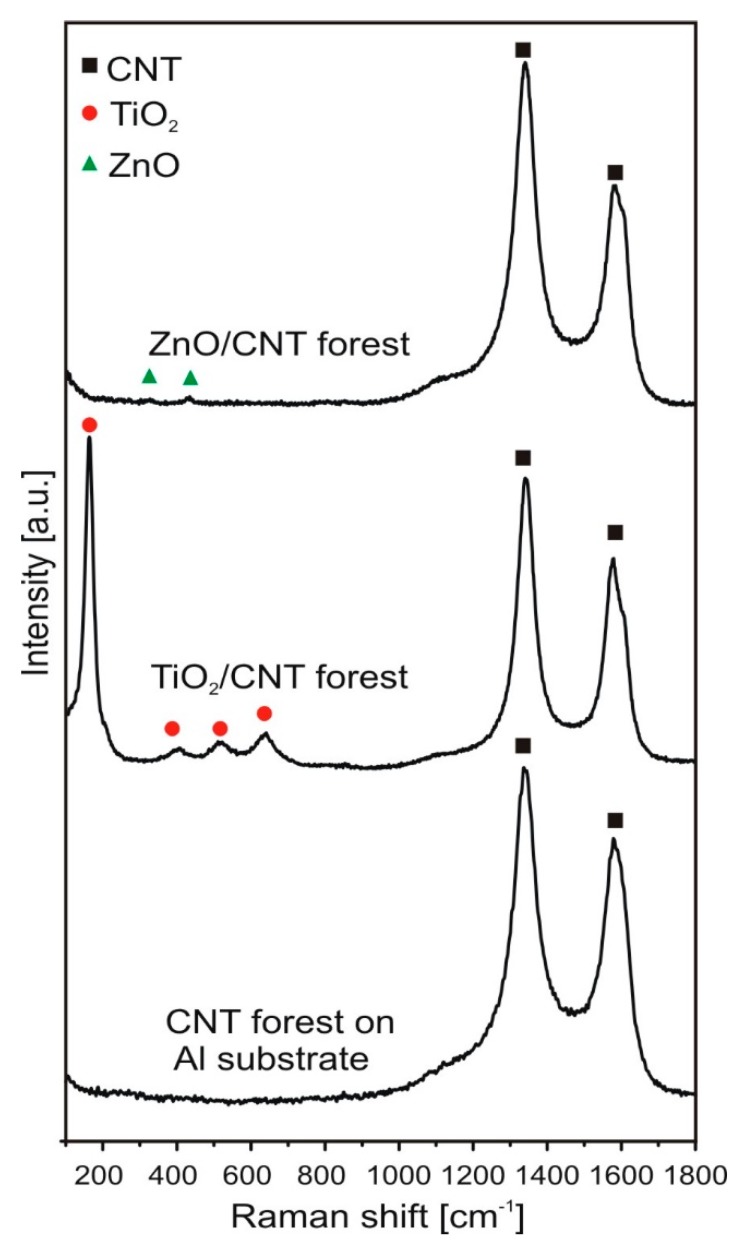
Raman spectra of the samples.

**Figure 5 materials-12-01095-f005:**
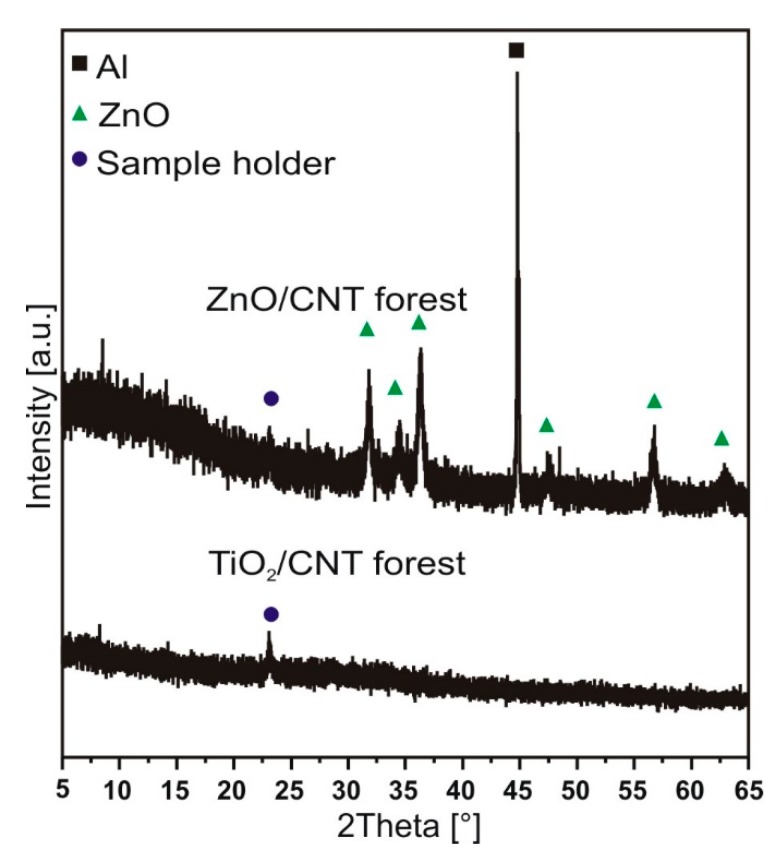
XRD diffractograms of the samples.

**Figure 6 materials-12-01095-f006:**
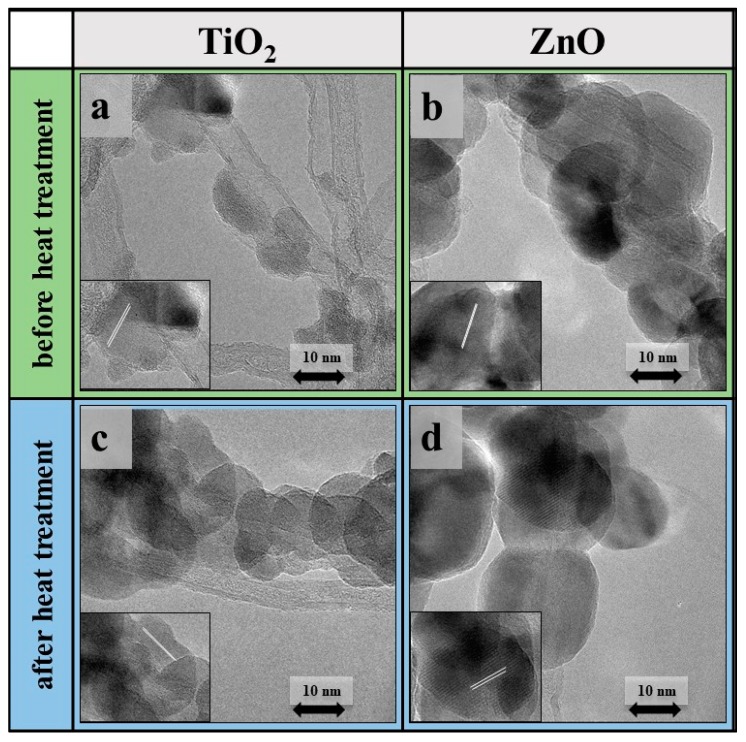
TEM images of TiO_2_ (**a**,**c**) and ZnO (**b**,**d**) coated CNT forests before (**a**,**b**) and after (**c**,**d**) heat treatment.

**Table 1 materials-12-01095-t001:** Composition of the samples from EDX spectra (b.HT: Before heat treatment; a.HT: After heat treatment).

Sample	Atomic %													
	**C**		**O**		Fe		Co		Ti		Cl		Zn	
	**b.HT**	**a.HT**	**b.HT**	**a.HT**	b.HT	a.HT	b.HT	a.HT	b.HT	a.HT	b.HT	a.HT	b.HT	a.HT
CNT	97.7	97.7	2.3	2.3	-	-	-	-	-	-	-	-	-	-
TiO_2_/CNT	51.3	23.1	35.5	61.6	0.1	0.0	0.1	0.0	12.9	13.8	0.1	1.5	-	-
ZnO/CNT	61.7	16.5	19.3	59.9	0.1	0.0	0.0	0.0	-	-	-	-	18.9	23.6

**Table 2 materials-12-01095-t002:** I_D_/I_G_ ratios, D and G shifts of the samples (b.HT: Before heat treatment; a.HT: After heat treatment).

Sample	I_D_/I_G_		D Shift [cm^−1^]		G Shift [cm^−1^]	
	**b.HT**	**a.HT**	b.HT	a.HT	b.HT	a.HT
CNT	1.26	1.26	1337.30	1337.30	1578.90	1578.90
TiO_2_/CNT forest	1.40	1.29	1341.69	1344.09	1578.90	1589.00
ZnO/CNT forest	1.57	1.08	1343.15	1346.97	1589.15	1585.15

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
