# Peer review of "Decoration of Vertically Aligned Carbon Nanotubes with Semiconductor Nanoparticles Using Atomic Layer Deposition"

_materials, 2019, doi:10.3390/ma12071095_

Reviewer 1 Report

The manuscript written by A. Szabó et al. is addressing the use of conventional ALD for preparing surface modifications for carbon nanotubes.  There are plenty of existing papers concerning ALD and nanotubes. Both of the TiO2 and ZnO preparation has been previously published so there is no novelty in ALD processing itself.  In addition there are quite limited amount of the ALD processing data, so strengthening of the manuscript should be carried out before it can be considered to be published in Materials. Either more deeper ALD process studies and/or additional sample characterization would make the paper much more interesting to the audience.

Some spell checking and some improvement on the text fluency should also be considered.

 Some specific comments:

Abstract: r. 33-34. It’s a little bit too positive to say that ALD is the best solution for coating VACNTs (albeit ALD has plenty of good features).

r. 131 what is the film thickness/morphology on VACNTs versus conventional substrates. How the ALD growth is proceeding?

r.162. More detailed information should be disclosed. What is the film uniformity on z-axis? What is total surface area to be coated? Is the ALD pulse time enough for uniform coating, when the surface saturation is obtained?  Is there any difference in ZnO and TiO2 depositions?

r 173. What could be the chemical reason for the loss of the carbon during the annealing? Could oxide films  (or -OH impurites) release oxygen? Is the carbon loss depending on the film thikness of annealing time? Could FT-IR help to resolve possible reactions? This is perhaps the most interesting point of the manuscript and it should be addressed in detail.

r. 181 Differences ZnO and TiO2 amount can be partly explained from different ALD deposition rates, but making some correlation studies between films thickness (deposited material) vs. deposition cycles could tell if there is different incubation time before ALD growth starts?

Author Response

We would like to thank to the reviewers and editorial board that our manuscript was suggested to be published after a careful revision. We made the requested changes and answered the questions and the revision of the manuscript. (Revised manuscript in correction mode is attached to this file.) Thank you for your time and effort and we hope that now this work can be accepted Materials. With respect,

The Authors

 The manuscript written by A. Szabó et al. is addressing the use of conventional ALD for preparing surface modifications for carbon nanotubes.  There are plenty of existing papers concerning ALD and nanotubes. Both of the TiO2 and ZnO preparation has been previously published so there is no novelty in ALD processing itself.  In addition there are quite limited amount of the ALD processing data, so strengthening of the manuscript should be carried out before it can be considered to be published in Materials. Either more deeper ALD process studies and/or additional sample characterization would make the paper much more interesting to the audience.

Thank you for your remarks and suggestions which inspired us for adding further experiments and hopefully for the improvement of our manuscript as well. See more details below.

Some spell checking and some improvement on the text fluency should also be considered.

Text was improved (see modifications in correction mode).

 Some specific comments:

Abstract: r. 33-34. It’s a little bit too positive to say that ALD is the best solution for coating VACNTs (albeit ALD has plenty of good features).

The sentence was modified:

“It was attested that atomic layer deposition is probably the mostasuitable technique for the fabrication of semiconductor/vertically aligned carbon nanotubes composites.”

r. 131 what is the film thickness/morphology on VACNTs versus conventional substrates. How the ALD growth is proceeding?

Section 2.4. was expanded significantly in the hope that we provided required information. However, the study of detailed mechanism of layer’s growth points much beyond the aim of current paper. Based on experimental results we presume the classical Volmer-Weber type (island) growth model with seeding at CNT defect sites; hopefully, these results will be published in a following research paper very soon.

r.162. More detailed information should be disclosed. What is the film uniformity on z-axis? What is total surface area to be coated? Is the ALD pulse time enough for uniform coating, when the surface saturation is obtained?  Is there any difference in ZnO and TiO2 depositions?

In section 2.4. further details are provided about both ALD conditions and results of complementary studies with glass substrates arranged evenly on the entire plate of the reaction chamber with diameter of 200 mm (both the substrates and the plate were larger than CNT forest) are also given. These show horizontal (x,y-axes) uniformity. If the horizontal uniformity is proved, it is very probable that vertically the situation is the same (z-axis) following from the nature of ALD method.

“A typical surface area value for multiwalled carbon nanotubes is approx. 180 m2/g. The estimated surface of CNT to be coated was a few cm2.” sentences were added to section 2.3. It is evident that the total surface of CNT forest is lower than test surfaces applied. Moreover, during ALD long pulse-purge cycles were applied in order to allow precursors to reach inner surfaces. Beneq TFS-200-186 ALD thermal reactor was developed for fast and provident growth: the diameter of the laminar flow chamber is 200 mm, while its height is only 3 mm. Accordingly, EM images revealed that homogeneous decoration was successful.

The difference in ZnO and TiO2 deposition is described in the paper: from the recipe it can be followed that both temperature and different precursors can be responsible for the variance.

r 173. What could be the chemical reason for the loss of the carbon during the annealing? Could oxide films  (or -OH impurites) release oxygen? Is the carbon loss depending on the film thikness of annealing time? Could FT-IR help to resolve possible reactions? This is perhaps the most interesting point of the manuscript and it should be addressed in detail.

Yes, either oxide film or impurities can be the source for carbon loss. Unfortunately, it was not possible to detect correlation between annealing time and film thickness. It is hard to find proper tool for complete analysis, however, we believe that coherency of several observations authorizes us for a speculative model described in the manuscript.

r. 181 Differences ZnO and TiO2 amount can be partly explained from different ALD deposition rates, but making some correlation studies between films thickness (deposited material) vs. deposition cycles could tell if there is different incubation time before ALD growth starts?

Expanded experimental section provides further details about ALD process.

In accordance with our former results [25], difference between the nature of titania and ZnO can also be a plausible reason. Previous results revealed that titania generally form significantly smaller (primary) particles, however, with higher density, than ZnO which can cause a difference in layer thickness, too. In other words, for titania nucleation, while for ZnO crystal growth is faster. This tendency can be also followed in the intensity of XRD peaks. These questions are really fascinating, but this study would require further deliberate experiments.

Reviewer 2 Report

 The paper reports the fabrication of vertically aligned carbon nanotubes decorated with TiO2 and ZnO semiconductor particles by atomic layer deposition. The nanotubes are characterized using several techniques such as scanning and transmission electron microscopy, energy-dispersive X-ray spectroscopy, Raman spectroscopy, and X-ray diffraction.

The paper is well written and reports a professionally conducted experimental work.  The topic is suitable for the journal.  I am in favor of its publication after a moderate revision:

 “The carbon nanotube forests are often used in electrical devices due to their electrical conducting properties and can be found in microelectromechanical devices [7], such as gas sensors [8], but can be also used in the preparation of nanocomposite systems [9].” The authors should add here that vertically aligned carbon nanotubes have been extensively used for field emission applications. See the following articles, which I suggest to add to the references: “A local field emission study of partially aligned carbon-nanotubes by atomic force microscope probe” Carbon 45(15):2957-2971, 2007 https://doi.org/10.1016/j.carbon.2007.09.049 and “Local probing of the field emission stability of vertically aligned multi-walled carbon nanotubesCarbon 47(4), pp. 1074-1080, 2009, https://doi.org/10.1016/j.carbon.2008.12.035 .

 “Due to the potential technological importance of semiconductor/CNT forest composites, there is an imperious demand to develop an effective synthesis process for its production.” The authors should be less generic here and mention a few key applications.

 In the caption of figure 2 specify what the insets of figure 2a and 2b represent.

 The ID/ IG ratio greater than 1 suggests that the CNTs used in this experiment are quite defective. The authors should comment on this and compare the quality of their CNTs with these usually reported in the literature. Is the low growth temperature contributing to the defectivity?

  “HR-TEM 225 images also revealed that the number of walls in carbon nanotubes were 4-5 on average and their diameter varied between 5-6 nm, while the average diameters of the TiO2 and ZnO particles were 25 nm and 30 nm, respectively.” Any estimation of the average length of the CNTs?

Author Response

We would like to thank to the reviewers and editorial board that our manuscript was suggested to be published after a careful revision. We made the requested changes and answered the questions and the revision of the manuscript. (Revised manuscript in correction mode is attached to this file.)  Thank you for your time and effort and we hope that now this work can be accepted Materials. With respect,

The Authors

The paper reports the fabrication of vertically aligned carbon nanotubes decorated with TiO2 and ZnO semiconductor particles by atomic layer deposition. The nanotubes are characterized using several techniques such as scanning and transmission electron microscopy, energy-dispersive X-ray spectroscopy, Raman spectroscopy, and X-ray diffraction.

The paper is well written and reports a professionally conducted experimental work.  The topic is suitable for the journal.  I am in favor of its publication after a moderate revision:

 “The carbon nanotube forests are often used in electrical devices due to their electrical conducting properties and can be found in microelectromechanical devices [7], such as gas sensors [8], but can be also used in the preparation of nanocomposite systems [9].” The authors should add here that vertically aligned carbon nanotubes have been extensively used for field emission applications. See the following articles, which I suggest to add to the references: “A local field emission study of partially aligned carbon-nanotubes by atomic force microscope probe” Carbon 45(15):2957-2971, 2007 https://doi.org/10.1016/j.carbon.2007.09.049 and “Local probing of the field emission stability of vertically aligned multi-walled carbon nanotubes” Carbon 47(4), pp. 1074-1080, 2009, https://doi.org/10.1016/j.carbon.2008.12.035 .

We appreciate your suggestion. Field emission applications and proposed references were added in lines 51-52.

 “Due to the potential technological importance of semiconductor/CNT forest composites, there is an imperious demand to develop an effective synthesis process for its production.” The authors should be less generic here and mention a few key applications.

Introduction part was completed with further applications and references. (see lines 60-61, 74-77, 94-95, 100-104)

 In the caption of figure 2 specify what the insets of figure 2a and 2b represent.

Figure caption was completed (lines 204-205) with element mapping data.

The ID/ IG ratio greater than 1 suggests that the CNTs used in this experiment are quite defective. The authors should comment on this and compare the quality of their CNTs with these usually reported in the literature. Is the low growth temperature contributing to the defectivity?

Yes, Al substrate applied for CNT forest growth required low CCVD temperature 640 ˚C which might be responsible for higher defectivity. However, this fact can be an advantage during ALD since defect sites are favorable for seeding. (lines 234-236)

  “HR-TEM 225 images also revealed that the number of walls in carbon nanotubes were 4-5 on average and their diameter varied between 5-6 nm, while the average diameters of the TiO2 and ZnO particles were 25 nm and 30 nm, respectively.” Any estimation of the average length of the CNTs?

To give further information, SEM image of as-prepared CNT forest was added (Fig. 1). The length of the CNTs is equal with the height of the forest.

Reviewer 3 Report

The paper by Szabo et al. describes the study of the verticaly alighen carbon nanotubes nanocomposites synthesis using atomic layer deposition. In my opinion the idea of the research is interesting, however there are some parts of the manuscript which should be rewritten. For instance, the title does not reflect the presented research. Now it looks like author used ALD to produce CNT. Second, introduction should be definitely approved. It is not clear the application of such nanocomposites and there is no overview on the current status of this topic.

As to Results and Discussion section, I would liek to ask why there are no peak, or they are too weak, of ZnO but we surely see (XRD) the presence of crystalline wurzite phase. What was the thickness of deposited metal oxides. Did authors compared these values with layers deposited on the plane surface. What are the perspectives of this nanocomposite and the synthesis method.

Author Response

We would like to thank to the reviewers and editorial board that our manuscript was suggested to be published after a careful revision. We made the requested changes and answered the questions and the revision of the manuscript. (Revised manuscript in correction mode is attached to this file.)  Thank you for your time and effort and we hope that now this work can be accepted Materials. With respect,

The Authors

The paper by Szabo et al. describes the study of the verticaly alighen carbon nanotubes nanocomposites synthesis using atomic layer deposition. In my opinion the idea of the research is interesting, however there are some parts of the manuscript which should be rewritten. For instance, the title does not reflect the presented research. Now it looks like author used ALD to produce CNT.

We appreciate your remark, the title was really not highly sophisticated. The new version is: Decoration of Vertically Aligned Carbon Nanotubes with Semiconductor Nanoparticles Using Atomic Layer Deposition

Second, introduction should be definitely approved. It is not clear the application of such nanocomposites and there is no overview on the current status of this topic.

Introduction part was completed with further applications and references. (see lines 51-52, 60-61, 74-77, 94-95, 100-104)

As to Results and Discussion section, I would liek to ask why there are no peak, or they are too weak, of ZnO but we surely see (XRD) the presence of crystalline wurzite phase. What was the thickness of deposited metal oxides.

Since the amount of the sample and the thickness of the inorganic layer on the surface of CNT was very low, the detection of different phases was very challenging. Therefore, we applied complementary techniques (Raman, XRD, HRTEM) for the proof of their presence. The thickness of the deposited metal oxides can be clearly seen from the HRTEM images.

Did authors compared these values with layers deposited on the plane surface.

Data for plane surface are provided. (Section 2.4.)

What are the perspectives of this nanocomposite and the synthesis method.

Potential applications are highlighted in the Introduction part.

Round  2

Reviewer 1 Report

Requested modifications and comments has been taken into account so the manuscript can be considered for publication in Materials.

Reviewer 2 Report

I appreciate the attention given to all my comments and suggestions. The authors have made changes and improvements in their manuscript. They have given convincing responses to the various questions and comments I had raised.

The revised version of the manuscript appears complete and technically sounder. The paper can be accepted for the publication in the current form. 

Reviewer 3 Report

I think authors have adressed all my main comments and remarks.